# Patch-Wise Automatic Segmentation for Real-Time PCB Inspection

## Abstract

Automated Optical Inspection (AOI) systems play a pivotal role in ensuring quality control during Printed Circuit Boards (PCB) manufacturing. However, the current AOI systems necessitate manual setting of the region of interest (ROI) for all components. To address this, we propose a patch-based preprocessing technique, dividing high-resolution PCB images into small $1024 \times 1024$ pixel patches and employing the YOLOv7 segmentation model for real-time component ROI segmentation. Our method consistently delivered high accuracy across various PCB components, irrespective of background color, and demonstrated robust performance even with complex structures containing small components. It achieved impressive outcomes, with an average IoU, F1 score, pixel accuracy, and mAP of 0.8889, 0.9401, 0.9961, and 0.8255, respectively. Specifically, utilizing Feature Pyramid Network (FPN) and Path Aggregation Network (PAN) in YOLOv7's multi-resolution processing allowed us to accurately segment PCB components of various sizes and process them in real-time. This study underscores the potential of automating real-time component ROI segmentation in the PCB manufacturing process to enhance production speed and quality control.

## 1 Introduction

Printed circuit boards (PCBs) are integral to nearly all electronic products, necessitating stringent quality and reliability standards. Ensuring the performance and safety of the final product involves identifying and correcting defects that may occur during PCB manufacturing. Automated Optical Inspection (AOI) is a key technique employed for these quality controls, such as inspecting PCBs during or after manufacturing to promptly identify defects. AOI systems utilize high-resolution cameras and advanced image processing algorithms to detect defects, including solder bridges, lifted pads, missing components, and misaligned parts (Hecht & Dishon, 1990).

The introduction of AOI has revolutionized the defect detection process, offering much faster and more accurate results compared to traditional manual inspection methods (Liao et al., 2018). This advancement significantly accelerated production speed and minimized the number of defective products shipped. Additionally, AOI operates in a non-contact manner, safeguarding PCBs and components from physical damage, thereby solidifying its status as an essential tool in modern PCB manufacturing (Czimmermann et al., 2020).

Despite the advancements made by AOI systems, most still require a manual setting process. This process involves defining a region of interest (ROI) for each PCB component to be inspected by the AOI system. Operators manually specify the position and size of each component to ensure the AOI system inspects them accurately. This procedure is exceedingly time-consuming, particularly for complex PCB designs that contain hundreds of components, each requiring precise ROI settings. This not only slows down the production line but also escalates labor costs and operator fatigue due to the high level of expertise and experience needed for the task. Furthermore, manual operations are prone to errors, resulting in inaccurate ROI settings, which in turn diminishes the AOI system's inspection accuracy and affects overall product quality.

To address these inefficiencies, there is a growing need for an automated ROI segmentation process. The automation of ROI setting can save time and costs, ensure consistent inspection quality, and reduce dependence on skilled labor. This study proposes a deep learning-based automated method

for partitioning PCB components, removing the need for manual instruction and enhancing the efficiency and reliability of the overall inspection process.

This study presents a novel preprocessing method to raise the learning efficiency and accuracy of deep learning models. Rather than processing entire PCB images, the images are cropped into 1024x1024 pixel-sized patches, effectively managing large data volumes and significantly improving model learning speed (Long et al., 2015).

Dividing PCB images into smaller patches offers several advantages. It concentrates the information within each patch, allowing the model to learn data more efficiently. This approach captures PCB details more effectively, enabling more accurate ROI settings and handling various component sizes and formats. Additionally, using small patches optimizes GPU memory usage, resolves memory shortage issues, and allows for larger batch sizes, which speeds up model convergence and reduces training time (Zhang et al., 2024). Training with small patches also enhances the model's ability to generalize, ensuring consistent performance across different PCB images.

With these goals, this study proposes a deep learning-based segmentation method for PCB AOI systems, incorporating preprocessing techniques to enhance learning efficiency and accuracy. This approach aims to elevate the automation level in PCB manufacturing processes and boost product quality. Ultimately, the study aspires to strengthen the competitiveness of the PCB manufacturing industry and contribute to the development of smarter manufacturing practices.

## 1.1 RELATED WORK

**Image Patch Division.** Research on dividing and processing images by patches has been used extensively to reduce memory usage and increase processing speed, especially in high-resolution image processing. These techniques can provide efficient and fast processing performance by dividing an image into smaller-sized patches and inputting each patch into the model rather than processing the entire high-resolution image.

An image patch division method was used to more efficiently detect retinal lesions in situations with a small amount of data by (Lam et al., 2018). When detecting lesions in retinal images, dividing and learning in small patch units rather than processing the entire image generated more training data, even with a small amount of data.

In the study by (Gao et al., 2013), a small target was effectively detected from a single infrared image. In infrared images, to solve the problem of small targets being easily hidden by the background or the features of the target being unclear, a patch-based model was proposed that divides an image into small patches and detects targets within each patch. It works in a way that highlights patches containing small targets and suppresses background noise, which helps to reduce computational complexity while maintaining high accuracy in detecting small-sized targets.

MegaDetectNet by (Wang et al., 2023b) detects targets from ultra-high resolution images by extracting patches of foreground images where the target exists, which avoids unnecessary image processing, maximizes computational efficiency, and significantly improves detection speed.

**Automated Solutions for PCB AOI.** Although the current industry recognizes the need for automated solutions for PCB AOI, several challenges still need to be solved to implement these solutions. (Anoop et al., 2015) compared various algorithms for detecting PCB defects, but the problem of setting ROIs with automatic segmentation has not been fully addressed.

Moreover, setting the ROI for numerous components in complex PCB designs is inefficient and increases the likelihood of errors. For instance, (Malge & Nadaf, 2014) utilized traditional image processing techniques to identify PCB defects, but these methods still depend on manual settings and lack automatic segmentation. Therefore, developing an automated segmentation solution to replace the manual process is essential to increase the efficiency of the PCB manufacturing process.

**Deep Learning-Based Segmentation in Different Domains.** Automatic image segmentation techniques based on deep learning have been widely studied and applied in various fields, including medical imaging, autonomous vehicles, and agriculture. In medical imaging, convolutional neural networks (CNNs), especially U-Net, have demonstrated strong performance in cell and tissue segmentation. (Ronneberger et al., 2015) introduced an image segmentation technique using U-Net that

performed well even with limited datasets. The model showed the ability to process medical images of various sizes and formats effectively.

In autonomous vehicles, Mask R-CNN is positioned as a powerful model that can simultaneously perform object detection and segmentation. (He et al., 2017) proposed a method to detect and segment objects from image data of autonomous vehicles using Mask R-CNN, which boasts high accuracy and efficiency. They have demonstrated that it can effectively segment vehicles, pedestrians, road signs, etc., in complex urban environments.

In agriculture, (Gao et al., 2020) proposed MMUU-Net for accurate segmentation of agricultural land using high-resolution satellite images, which can efficiently distinguish agricultural and non-agricultural areas and can be utilized in various fields such as agricultural management, crop production forecasting, and environmental monitoring. Such deep learning-based segmentation techniques can be equally applied to PCB inspection, especially useful for efficiently handling numerous complex components. Therefore, automated segmentation solutions are likely to be positioned as a pivotal technology to maximize the efficiency of PCB AOI systems and further improve the automation level of manufacturing processes.

## 2 METHODOLOGY

### 2.1 PATCH-WISE PREPROCESSING OF PCB IMAGES

Learning a model using an image of an entire PCB comes with several challenges due to its size and complexity. PCB images have high resolution and contain a wide variety of components, requiring substantial computing resources to process effectively. In particular, when using large datasets to train deep learning models, problems may occur due to resource limitations such as GPU memory. Thus, a preprocessing process was applied to partition PCB images into small patches of 1024 × 1024 pixels in order to maximize learning efficiency and model performance.

In this study, we proposed a preprocessing method to divide PCB images into 1024 × 1024 pixel size patches for processing. Learning a model using entire PCB images results in excessive memory usage and reduces learning efficiency due to high resolution and complexity. PCB images are usually 10,000 x 15,000 pixels and can be up to 25,000 x 30,000 pixels. On the other hand, components are small in size, ranging from 40 x 80 pixels to 1,000 x 1,000 pixels, significantly smaller than the overall board size, making it highly inefficient to use the entire board images for training.

To solve this, we divided all images into small patches of 1024 × 1024 pixel size based on the bottom left to the top right of the PCB, as shown in Figure 1. If the width and length of the PCB are not divided by 1024, the remaining parts on the right and top edges are filled with black to make the 1024 x 1024 pixel size patches. This preprocessing method is advantageous because, unlike domains such as medical images where targets or lesions are concentrated at specific positions, PCB components are spread throughout the board, so all components are evenly distributed in patches. Each patch was processed in a separate learning unit, and the model was trained to segment components in the patch accurately. After learning about the patches divided in this way, the results were output in a patch state and recombined into one PCB image. All components segmented in each patch were reconstructed across the entire PCB. This made it possible to segment all materials on the PCB by type.

This preprocessing optimizes memory usage during learning and inference while providing uniform visual information that can be segmented globally without missing small objects. Moreover, since each patch is processed independently, it has a fast training speed and can efficiently process large PCB images without memory constraints.

### 2.2 YOLOv7 SEGMENTATION

In this study, we adopted the YOLOv7 segmentation model to automate ROI segmentation in PCB AOI systems (Wang et al., 2023a). PCB AOI systems are essential for real-time analysis of PCB images, and YOLOv7 is known for its impressive speed, delivering high Frames Per Second (FPS) to process multiple images per second efficiently. YOLOv7 achieves 30 FPS or higher on standard

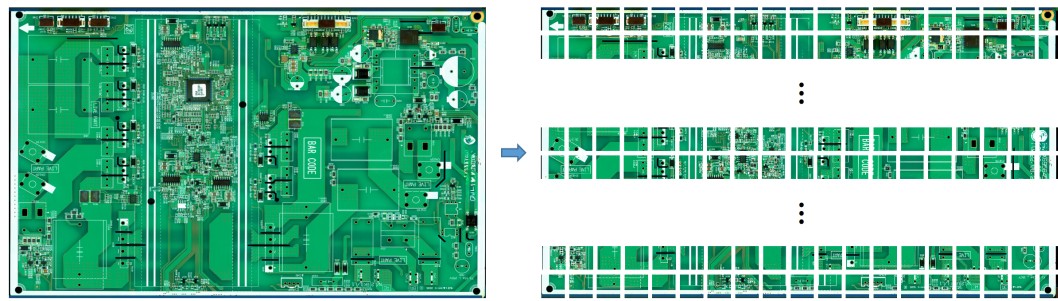

Figure 1: Patch-wise preprocessing. An example of an 18242 x 14782 pixel size PCB image divided into 1024 x 1024 pixel size patches from the bottom left corner to the top right corner.

GPU setups, outpacing alternatives such as YOLOv4 and Faster R-CNN, which makes it ideal for high-speed PCB inspection.

Since this study is to draw accurate ROI for all components, a segmentation head was used instead of a detection head so that output could be in pixel-wise class labels instead of bounding boxes. The segmentation head involves a series of upsampling layers and convolutional layers to increase the resolution of the feature maps and predict the class of each pixel. In addition, YOLOv7's segmentation capability is particularly noteworthy due to its Enhanced-Efficient Layer Aggregation Network (E-ELAN) architecture. It is a cutting-edge architecture in object detection that draws inspiration from the successful foundations laid by Feature Pyramid Network (FPN) and Path Aggregation Network (PANet). While building upon the strengths of these predecessors, E-ELAN introduces novel techniques to streamline feature aggregation and enhance performance. By efficiently routing information across feature levels and adaptively fusing features, E-ELAN demonstrates a remarkable ability to handle objects of varying sizes and capture fine-grained details. This architectural innovation solidifies E-ELAN's position as a leading contender in the field of object detection and makes the entire network simply look like Figure 2.

Furthermore, YOLOv7 enhances its efficiency through lightweight operations and the Bag-of-Freebies technique. Unlike traditional deep learning models that rely on complex calculations, YOLOv7 minimizes computational overhead while maintaining accuracy by integrating various methods that improve performance without requiring additional training. This efficiency makes YOLOv7 well-suited for real-time PCB AOI applications.

In this experiment, we utilized Stochastic Gradient Descent (SGD) as our optimization algorithm with an initial learning rate of 0.01, a momentum of 0.937, a weight decay of 5e-4, and a maximum learning rate of 0.1 for the OneCycleLR scheduler. Weight decay is a regularization technique that helps prevent overfitting by penalizing large weights. The combination of these optimization parameters, along with the OneCycleLR learning rate scheduler, contributed to the model's effective convergence and generalization performance.

In addition, we utilized the Binary Cross Entropy with Logits Loss (BCEWithLogitsLoss) to estimate the likelihood of each pixel being associated with a specific class. The BCEWithLogitsLoss is particularly effective in clearly delineating the boundary between two classes and is commonly employed as a loss function, especially in split operations. To optimize GPU memory usage, the batch size was set to 8 and the epoch to 500. Given the high-resolution nature of PCB images, we chose a relatively small batch size to aid in memory management.

## 3 EXPERIMENTS

### 3.1 DATASET

In this study, 67 PCB images were collected directly from the AOI system, and their patches were manually labeled. Instead of labeling all components in one class, ten labeling classes were composed, and each class was defined based on the main electronic elements in the PCB. The labeled classes consisted of lead, pad, chip, resistor, capacitor, diode, integrated circuit (IC), connector,

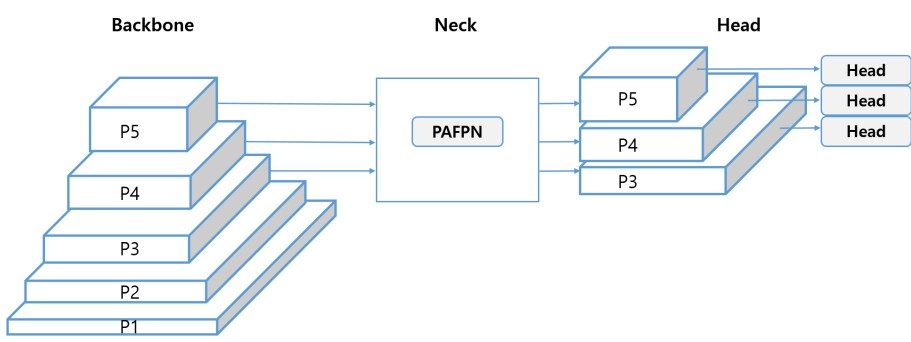

Figure 2: Simplified yolov7 model

light-emitting diode (LED), and coil. Each component plays an essential role in the electrical performance and operation of the product, so individual labeling is required.

The work of (Li et al., 2020) focused on the pixel-by-pixel division of the location that a specific component occupies in the actual PCB and its material in a way that individually labels the materials in the PCB. This aimed for accurate location detection and division of each item rather than a specific classification of the type of item. Although this method is advantageous in effectively capturing the location of individual components in a PCB, it has limitations in reflecting the functional differences between components because it treats all materials in the same way without distinguishing between material types.

Instead of the comprehensive labeling commonly used in previous studies, this study classifies each PCB component separately, which allows more accurate segmentation results to be derived. For example, resistors regulate current flow, and capacitors store charge. Such functionally essential elements need to be labeled individually to ensure the stability of the operation of electronic devices. In addition, components in a PCB vary in size and shape, so classes must be subdivided.

In addition, the labeling work was done manually by experienced AOI inspection operators to minimize possible errors in the labeling process. Repeated reviews were conducted to ensure that the class of each component was classified correctly and whether the labeled areas matched the actual components. This was a crucial step to improve the quality of the training data and enable the model to make more accurate predictions.

### 3.2 TRAINING

In this study, we leveraged the NVIDIA A100 80GB PCIe GPU. The A100 GPU is optimized for processing large data and high-resolution images, enabling optimal performance, particularly for tasks that demand substantial memory resources. With its 80GB memory capacity, we effectively mitigated memory constraints while supporting high-resolution image processing and large batch sizes.

Of the total 67 PCB images, 52 were split for training, 10 for validation, and 5 for testing. We specifically selected five different test PCBs with varying background colors to assess the performance across different backgrounds. The data split was carefully planned to ensure a diverse representation of components. Given the diverse material types of PCBs, 70% of all PCBs were utilized for training.

## 4 RESULTS

In this experiment, five PCBs were selected as the test dataset to evaluate the model's performance under various conditions. These boards are large in size, contain hundreds of components, ranging from 286 to 956, and each has a different background color - blue, green, brown-green, white, and black. The diverse selection of background colors was deliberate to ensure the model's consistent ability to detect components regardless of background variations.

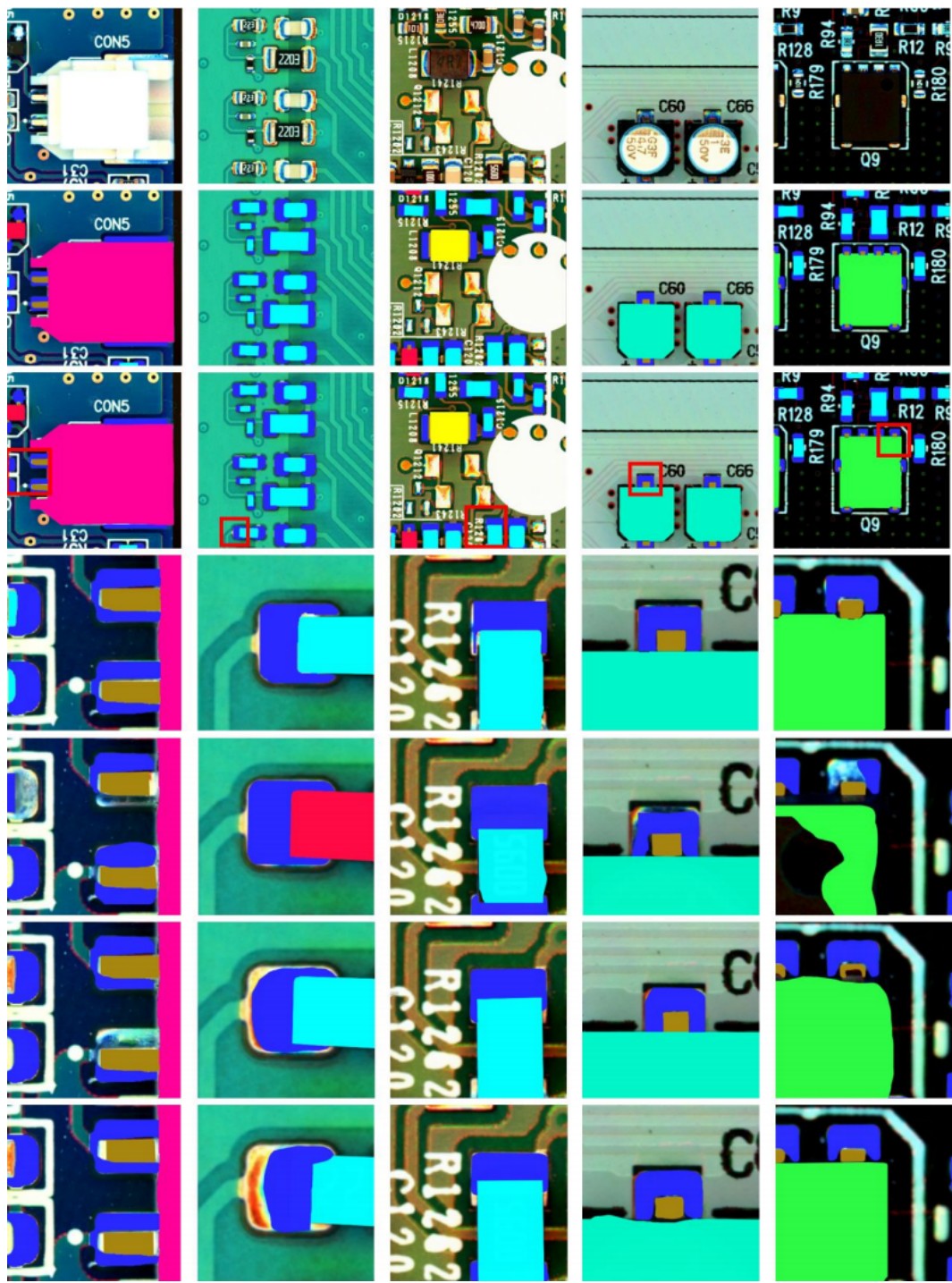

Figure 3: From left to right column: blue, green, brown-green, white and block boards; from top to bottom row: original image, ground truth, prediction, zoomed-in view of prediction, and the corresponding results from DeepLabv3+(Chen et al., 2018), Mask R-CNN(He et al., 2017), and YOLACT(Bolya et al., 2019). Areas highlighted by red boxes in the prediction are critical regions of interest and zoomed in for a detailed evaluation. Each class is color-coded as follows: leads in tan, pads in blue, chips in cyan, connectors in pink, coils in yellow, resistors in red, ICs in green, capacitors in turquoise, LEDs in black, and diodes in purple.

In PCB manufacturing environments, various background colors are used, and each one can affect the lighting, image quality, and inspection. Therefore, it was crucial to assess the model's performance by testing in such various conditions. In particular, blue and green are common PCB background colors, making it challenging to distinguish components against these backgrounds. Additionally, black and white could affect segmentation performance due to contrast.

The proposed preprocessing method and the use of YOLOv7 segmentation model in this study enabled segmentation of PCB components even under various background conditions, as shown in Figure 3. Using the test dataset, the model successfully segmented components with diverse colors and shapes. Segmentation of components with irregular shapes and colors, such as connectors, can be particularly challenging. Connectors often have irregular shapes and various colors, increasing the likelihood of errors in an automated segmentation system. However, the YOLOv7 segmentation model used in this study adeptly identified and segmented the intricate shapes and irregular colors of connectors. This confirmed that the model can handle components of various shapes and colors.

Moreover, in the case of black ICs on the black board, the last column of Figure 3, the model showed exceptional performance even when there was minimal color contrast between the board and the component, which made it difficult to distinguish the boundaries. While segmented components in a generally low-contrast setting may be arduous, this study overcame these challenges and segmented the boundaries of black ICs on the black board. In the case of DeepLabv3+, it was observed that more than 70% of the IC could not be segmented. The model struggled to distinguish the boundary between the black background and the black IC, leading to incomplete segmentation of the IC area. Furthermore, the leads and pads were not segmented completely, highlighting the model's limitations in delineating boundaries in low-color contrast environments. Similarly, Mask R-CNN also faced challenges in accurately detecting the boundary between the background and the IC, resulting in consistently irregular results. This can be attributed to a performance drop in boundary recognition, leading to incomplete segmentation of the leads and pads. While Mask R-CNN excels in detecting object shapes, its limitations are evident in low-contrast scenarios. On the other hand, YOLACT demonstrated relatively good performance in segmenting the IC but struggled to detect the dark parts of the pads accurately. Despite its fast speed and good performance, YOLACT exhibited performance degradation in certain small and dark areas, such as pads.

Chips of different colors and sizes were also accurately segmented, as shown in the second column of Figure 3. Given the diversity in chip sizes and colors, performing uniform segmentation presented an achievement. Additionally, the segmentation performance of leads and pads may vary depending on the lighting reflection and board color. However, they were all segmented accurately despite their sizes and lighting conditions.

In contrast, DeepLabv3+ segmented the pads relatively accurately but misidentified the chip as a diode. The distinction between chips and diodes was unclear, and confusion occurred due to similar color and shape, highlighting its limitation in boundary distinction. Mask R-CNN tended only to detect parts with different colors in the pad segmentation. It was unable to segment the entire pad, but only segmented parts with high illumination values, and as a result was unable to recognize the entire pad. Lastly, YOLACT misidentified polar parts on the side of the chip as a pad, resulting in the incorrect segmentation of the chip and incomplete segmentation of the pad. Overall, these results demonstrate specific limitations of each model in segmenting chips and pads.

The model performance is quantitatively evaluated using metrics in Table 1, with an average IoU, F1 score, pixel accuracy, and mAP of 0.8889, 0.9401, 0.9961, and 0.8255, respectively. This suggests that the segmentation of PCB components showed high accuracy overall. In particular, the IoU

Table 1: Evaluation metric values and their average values of the test dataset.

| Board color | Blue | Green | Brown-green | White | Black | Average |
|---|---|---|---|---|---|---|
| IoU | 0.9133 | 0.8692 | 0.8823 | 0.8851 | 0.8948 | 0.8889 |
| F1 Score | 0.9539 | 0.9281 | 0.9364 | 0.9337 | 0.9485 | 0.9401 |
| Pixel Accuracy | 0.9958 | 0.9975 | 0.9960 | 0.9957 | 0.9954 | 0.9961 |
| mAP | 0.8698 | 0.7917 | 0.8033 | 0.8345 | 0.8281 | 0.8255 |

Table 2: Comparison of the average evaluation metric value and inference time in milliseconds for DeepLabv3+, Mask R-CNN, YOLACT, and YOLOv7 on the test dataset.

|                     | DeepLabv3+ | Mask R-CNN | YOLACT | YOLOv7    |
| ------------------- | ---------- | ---------- | ------ | --------- |
| IoU                 | 0.8178     | 0.8309     | 0.8556 | **0.8889** |
| F1 Score            | 0.8976     | 0.9001     | 0.9113 | **0.9401** |
| Pixel Accuracy      | 0.9283     | 0.9297     | 0.9488 | **0.9961** |
| mAP                 | 0.6919     | 0.7113     | 0.7151 | **0.8255** |
| Inference time (ms) | 60-80      | 45-60      | 20-35  | **10-20** |

was over 0.86, the F1 score was over 0.92, and the pixel accuracy was over 0.99 for all test PCBs, confirming that the model performed consistently in various background conditions.

Compared with DeepLabv3+, Mask R-CNN, and YOLACT, YOLOv7 recorded the highest performance across all metrics, as shown in Table 2. It achieved an overall average IoU of 0.8889, an F1 score of 0.9401, a Pixel Accuracy of 0.9961, and an mAP of 0.8255. These results demonstrate YOLOv7's exceptional capability in accurately segmenting PCB components, and effectively handling components of various sizes and shapes. This makes it highly suitable for real-world environments where precision and consistent performance are crucial in PCB AOI systems.

Additionally, YOLOv7 showed the fastest inference speed at 10-20ms. DeepLabv3+ recorded inference times of 60-80ms, Mask R-CNN recorded 45-60ms, and YOLACT recorded 20-35ms, confirming that YOLOv7 is the best model for real-time processing. In PCB manufacturing, instant inspection and quality control are essential to keep pace with rapid production. Hence, YOLOv7's fast processing speed and high accuracy make it the optimal choice for real-time PCB inspection, rendering it highly valuable for application in manufacturing facilities.

## 5 DISCUSSION

In the proposed patch-wise learning, since images are divided into a constant size of 1024 × 1024 pixels and processed, it was observed that small components could not be recognized when they fell on the patch boundary, or only a part of the object was detected. This issue occurred frequently, especially for small components or patch boundaries on the PCB, which can be evaluated as one of the inherent limitations of patch-wise learning. Hence, the average IoU was relatively low at 0.8889. Since tiny objects span the boundary in the patch-wise processing, they are undetected or partially detected.

In addition, while the body part of the component was segmented with high accuracy, smaller elements, such as leads and pads, proved to be relatively difficult to segment, as shown in the third column-fourth row of Figure 3 that two small parts of pads are undetected. This is interpreted as being due to the fact that elements such as leads and pads are small compared to the body, and their location on the PCB posed difficulties to model learning. Thus, relatively poor boundaries and small sizes are other factors that lead to a low IoU value on average.

## 6 CONCLUSION

In this study, we proposed a patch-wise preprocessing method to partition PCB images into patches for efficient component ROI setting in PCB AOI systems. By dividing PCB images into 1024 × 1024 pixel patches and processing them, we achieved significantly accurate segmentation results for all components on PCBs. Furthermore, by utilizing YOLOv7 segmentation model, we solved the memory usage problem that occurs when processing high-resolution images and enabled faster and more efficient inference. The proposed patch-wise learning consistently demonstrated strong performance with PCB images of varying background colors and sizes and enabled accurate segmentation across different components such as resistors, capacitors, and ICs. However, a limit was

found for small targets such as leads and pads across the patch boundaries, with a relative decrease in segmentation performance.

In future research, finer patch sizes or multi-resolution approaches can be considered to overcome these limitations and improve the segmentation performance of small components. Moreover, further experiments in various PCB design and manufacturing environments are necessary to validate the widespread applicability of the proposed patch-wise method in industrial sites. This study highlights the potential of an effective preprocessing method with YOLOv7 segmentation model for automatic ROI setting in the PCB AOI system, aiming to enhance the speed and accuracy of inspections in the PCB manufacturing process.

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
