# OpenReview forum: "Patch-Wise Automatic Segmentation for Real-Time PCB Inspection"
_ICLR.cc/2025/Conference — ICLR 2025 Conference Withdrawn Submission_

### Official Review · Reviewer_SiFi · 2024-10-18

**Soundness:** 1
**Presentation:** 3
**Contribution:** 1
**Rating:** 1
**Confidence:** 4

**Summary:**

This work presents a patch-wise preprocessing step, followed by a segmentation model of PCB components. The segmentation model is YOLOv7, which was tuned on a dataset that was annotated by experts.

**Strengths:**

This work presents an algorithm that seem to successfully segment the required PCB components.
They provide few metrics and report on few ablation experiments that compare different segmentation backbones.
The paper is clearly written and is easy to follow.

**Weaknesses:**

1. The authors do not compare to other PCB segmentation works, but rather try out different segmentation modules on their dataset. Since this work is not the first to offer segmentation on PCB images, it needs to be compared to the works to be evaluated properly.
Here is only a short list of such segmentation works for PCB images:
https://ieeexplore.ieee.org/stamp/stamp.jsp?tp=&arnumber=9721901

https://www.sciencedirect.com/science/article/pii/S0957417423005316?casa_token=Shzl_BwIRDAAAAAA:_m9fw8RWrpezvvyzvQE7aXALKORhVHhVvsPpQnzQ9xKXtWE-_BEmHwXCupJxYw6rouNlWw4_

https://ieeexplore.ieee.org/stamp/stamp.jsp?arnumber=6622859&casa_token=W8IwkaZqz94AAAAA:53AiF7a8-rUUKlCJJEqF-2QcbAQIWPhIqsyTq4Z_5uC0_6tpSp6QdeUtrmsAiWp2CO7Odl42

2. I wasn't able to find anything novel in this work. Applying sliding-window as a pre-processing step is obviously not new (in terms of general images as well as on PCB images), and is presented in this work as something novel and one of the contributions of the work. Utilizing another known architecture and tuning it to a private dataset also does not seem novel to me.

There is lots of text redundancy specifically in the Methodology section. The patch-wise preprocessing step subsection, could have been presented as "we applied sliding-window with the following parameters ..." and that differently from other domains, all patches can be used since all of them contain important PCB components (as you stated). The YOLOv7 Segmentation subsection feels like another related work section. It looks like another survey of that work.

**Questions:**

Will the code or dataset be public?
If so, and if the dataset is unique and of interest to the community, you should consider presenting it as another contribution.

---

### Official Review · Reviewer_Ws1A · 2024-10-29

**Soundness:** 2
**Presentation:** 2
**Contribution:** 2
**Rating:** 3
**Confidence:** 4

**Summary:**

This paper proposes a patch-wise automatic method for real-time PCB inspection, proposing a patch-based preprocessing method to divide the high-resolution PCB images into small patches. Second, YOLOv7 with a segmentation head is used to perform ROI segmentation for PCB components. The method demonstrates good performance on the collected PCB dataset.

**Strengths:**

1. This method uses a patch-based approach to handle high-resolution PCB images.
2. This paper reports strong performance across various PCB components.

**Weaknesses:**

1. This paper divides high-resolution PCB images into uniform 1024 × 1024 fixed-resolution patches. This method has already been applied in many computer vision tasks, such as object detection in remote sensing, and authors should claim the difference and advance of the proposed method.

2. The paper applies YOLOv7 with a segmentation head directly for segmentation without any significant improvement for PCB images.

3. The experiments are not comprehensive enough. The experimental section should include more recent comparison methods, as well as ablation studies, such as variations in patch size and slicing methods.

4. The multiplication symbol is not used correctly. For instance, in line 138.

**Questions:**

See above.

---

### Official Review · Reviewer_xVMY · 2024-10-31

**Soundness:** 2
**Presentation:** 2
**Contribution:** 2
**Rating:** 3
**Confidence:** 5

**Summary:**

Aimed at improving Automated Optical Inspection in PCB manufacturing, this study presents a patch-based preprocessing technique and employs YOLO-v7. The results demonstrate some performance improvements.

**Strengths:**

The authors collected the data and achieved commendable results through their claimed approaches, demonstrating a well-executed and relatively comprehensive engineering project.

**Weaknesses:**

1. PCB defect detection is a thoroughly explored area, but the authors have not sufficiently reviewed related work, and their identification of the challenges within PCB defect detection is unclear. Numerous studies focus on this topic, with many published in industrial intelligence journals such as TII, TIE, TIM, TSM, JMS, JMP, JIM, IJAMT, etc.
2. The method lacks innovation, as the use of image patching combined with YOLO detection is already a well-established engineering solution. The novelty could be weaker than papers from the mentioned journals.
3. The experimental validation does not utilize existing publicly available PCB defect detection datasets, which limits the robustness of the findings.
4. Overall, the quality doesn't match ICLR. Suggesting submitting to industry-related journals after careful improvements.

**Questions:**

1. Will the authors plan to release the dataset? The new dataset could also be a contribution. Of course it would be better if a larger dataset.
2. The authors could further improve the structure and expressions. eg. the structure of Results Section could be improved, the Inference time could be replaced by more common frames per seconds (fps), etc.

---

### Official Review · Reviewer_SxGY · 2024-11-01

**Soundness:** 2
**Presentation:** 2
**Contribution:** 1
**Rating:** 3
**Confidence:** 4

**Summary:**

This paper addresses challenges in Automated Optical Inspection (AOI) systems used in Printed Circuit Board (PCB) manufacturing. These systems traditionally require manual setting of the region of interest (ROI) for each component, a process that is time-consuming and prone to errors.

**Strengths:**

+ This paper presents a straightforward approach for detecting defects in high-resolution PCB images.
+ The paper is easy to read, and the experimental results effectively demonstrate the method's performance.

**Weaknesses:**

# Movitation
- The paper lacks a clear contribution, as the methodology remains closely aligned with YOLOv7, showing limited novelty. Additionally, splitting and patching high-resolution images into smaller tiles is a widely used, conventional technique, which further limits the originality of the approach.
- Given the goal of processing high-resolution images, the authors should provide stronger justification for selecting YOLOv7 as the core model. It would be beneficial to discuss why this choice is preferable over more recent or specialized methods, especially considering recent advancements.

# Experiment
- The comparison benchmarks in the experimental section are outdated. Including evaluations against state-of-the-art transformer-based detectors or segmentors, such as DETR, SAM or DINOv2 is essential.
- Moreover, comparisons with recent methods specifically targeting small object detection should be added to SOTA table.

**Questions:**

- What is the difference between your approach and slicing the image directly into patches for detection?
- Please add a comparison with popular transformer-based methods.
- What is the computational resource consumption of the method, such as FPS and paramters?

---

### Note · Authors · 2024-11-19

I have read and agree with the venue's withdrawal policy on behalf of myself and my co-authors.